# On Decoder Ties for the Binary Symmetric Channel with Arbitrarily Distributed Input

**DOI:** 10.3390/e25040668

**Published:** 2023-04-16

**Authors:** Ling-Hua Chang, Po-Ning Chen, Fady Alajaji

**Affiliations:** 1Department of Electrical Engineering, Yuan Ze University, Taoyuan 32003, Taiwan; 2Institute of Communications Engineering, National Yang-Ming Chiao-Tung University, Taipei 112304, Taiwan; poningchen@nycu.edu.tw; 3Department of Mathematics and Statistics, Queen’s University, Kingston, ON K7L 3N6, Canada; fa@queensu.ca

**Keywords:** binary symmetric channel, block codes, non-uniformly distributed channel inputs, joint source-channel coding, maximum a posteriori (MAP) decoding, decoder ties, error probability, error exponent

## Abstract

The error probability of block codes sent under a non-uniform input distribution over the memoryless binary symmetric channel (BSC) and decoded via the maximum a posteriori (MAP) decoding rule is investigated. It is proved that the ratio of the probability of MAP decoder ties to the probability of error grows most linearly in blocklength when no MAP decoding ties occur, thus showing that decoder ties do not affect the code’s error exponent. This result generalizes a similar recent result shown for the case of block codes transmitted over the BSC under a uniform input distribution.

## 1. Introduction

Consider the classical channel coding context, where we send a block code through the memoryless binary symmetric channel (BSC) with crossover probability 0<p<1/2. Given a sequence of binary codes ${\left\{C_{n}\right\}}_{n\geq 1}$ with *n* being the blocklength, we denote the sequence of corresponding minimal probabilities of decoding error under *maximum a posteriori (MAP) decoding* by ${\left\{a_{n}\right\}}_{n\geq 1}$. The following result was recently shown in [1] when the channel input selects codewords from $C_{n}$ according to a uniform distribution.

**Theorem** **1**([1]). *For any sequence of codes ${\left\{C_{n}\right\}}_{n\geq 1}$ of blocklength n and size $|C_{n}|=M$ with $C_{n}\subseteq {\left\{0,1\right\}}^{n}$, sent over the BSC with crossover probability 0<p<1/2 under a uniform channel input distribution over $C_{n}$, its minimum probability of decoding error $a_{n}$ satisfies*
(1)$$b_{n}\leq a_{n}\leq \left(1+\frac{\left(1-p\right)}{p}n\right)b_{n},$$*where*
(2)$$\begin{array}{cc} b_{n} & =P_{X^{n},Y^{n}}\left\{\left(x^{n},y^{n}\right)\in X^{n}\times Y^{n}:P_{X^{n}|Y^{n}}\left(x^{n}|y^{n}\right)<\max_{u^{n}\in C_{n}\backslash \{x^{n}\}}P_{X^{n}|Y^{n}}\left(u^{n}|y^{n}\right)\right\}, \end{array}$$
*where $P_{X^{n},Y^{n}}$ is the joint input–output distribution that $X^{n}=\left(X_{1},X_{2},\ldots ,X_{n}\right)\in X^{n}≜{\left\{0,1\right\}}^{n}$ is sent over the BSC (via n uses) and $Y^{n}=\left(Y_{1},Y_{2},\ldots ,Y_{n}\right)\in Y^{n}≜{\left\{0,1\right\}}^{n}$ is received.*

Noting that $b_{n}$ in (2) is the probability that a decoding error occurs without inducing decoder ties (which occur when two or more codewords in $C_{n}$ are identified by the decoder as the estimated transmitted codeword; i.e., when more than one codeword in $C_{n}$ maximize $P_{X^{n}|Y^{n}}\left(\cdot |y^{n}\right)$ for a given received word $y^{n}$), the above result in (1) directly implies that decoder ties do not affect the error exponent of $a_{n}$. The error exponent or reliability function of a block coding communication system represents the largest rate of exponential decay of the system’s probability of decoding error as the coding blocklength grows to infinity (e.g., see [2,3,4,5,6,7,8,9,10,11,12,13,14]).

It is known that uniformly distributed data achieves the largest entropy rate and leaves no room for data compression. Thus, ideally compressed data should exhibit uniform distribution for all blocklengths *n*. However, this setting is often impractical due to the sub-optimality of the implemented data compression schemes. Instead, we generally have non-uniformly distributed data after compression in the form of residual redundancy such as in speech or image coding (e.g., [15,16]). Furthermore, one may have a compressed source that can be divided into several groups, within each of which the symbols are equally probable. Decoder ties can thus occur with respect to two (or more) codewords corresponding to symbols within the same group.

In this paper, we consider a non-uniform prior distribution over $C_{n}$ and prove that decoder ties, under optimal MAP decoding, still have linear and hence sub-exponential impact on the error probability $a_{n}$, thus extending Theorem 1 established for the case of a uniform prior distribution over $C_{n}$. Since our problem falls within the general framework of joint source-channel coding for point-to-point communication systems, we refer the reader to [14,15,16,17,18,19,20,21] (Section 4.6) and the references therein for theoretical studies on this subject as well as practical designs that outperform separate source and channel coding under complexity or delay constraints.

The proof technique used in [1] to show (1) above is based on the observation that there are two types of decoding errors. One is that the received tuple at the channel output induces no decoder ties but the corresponding decoder decision is wrong. The other is that the received tuple at the channel output causes a decoder tie, but the decoder picks the wrong codeword. As a result, the MAP error probability $a_{n}$ can be upper bounded by the sum of two terms, $b_{n}$ and $\delta_{n}$, where $b_{n}$ is the probability of the first type of decoding errors as given in (2), and $\delta_{n}$ is the probability of decoder ties regardless of whether the tie breaker misses the correct codeword or not. Under the assumption that the channel input is uniformly distributed over block code $C_{n}$ for each blocklength *n* and an arbitrary sequence of codes ${\left\{C_{n}\right\}}_{n\geq 1}$, it was shown in [1] that flipping a properly selected bit component of the channel output that causes a decoder tie can produce a unique channel output that leads to the first type of decoding errors. An analysis of this bit-flipping manipulation shows that the ratio $\delta_{n}/b_{n}$ grows at most linearly in *n* and hence yields the upper bound in (1). However, this flipping technique no longer works when non-uniform channel inputs are considered. To tackle this problem, we judiciously separate the channel output tuples that induce decoder ties into two groups, one group consisting of output tuples that do not fulfill the above flipping manipulation property and the other group composed of the remaining output tuples (i.e., the complement group). We then show that the probability of the former group is upper bounded by that of the latter group, and therefore $\delta_{n}/b_{n}$ remains growing at most linearly in blocklength *n* under arbitrary channel input statistics. Note that the group that fails the flipping property is an empty set when channel input is uniformly distributed over $C_{n}$, thereby making the result of Theorem 1 a special case of the extended result in this paper. The rest of the paper is organized as follows. Section 2 presents the main result and highlights the key steps of the proof to facilitate its understanding. The proof is then provided in full detail, along with illustrative examples, in Section 3 and Appendix A and Appendix B. Finally, conclusions and future directions are given in Section 4.

Throughout the paper, we denote [M]≜{1,2,…,M} for positive integer *M* and set $d(x^{n},y^{n}|S)$ to be the Hamming distance between *n*-tuples $x^{n}=\left(x_{1},x_{2},\ldots ,x_{n}\right)$ and $y^{n}=\left(y_{1},y_{2},\ldots ,y_{n}\right)$ with the indices of the tuples restricted to S⊆[n]. By convention, we set $d(x^{n},y^{n}|S)=0$ when S=∅ and use $d(x^{n},y^{n})$ to represent $d(x^{n},y^{n}|\left[n\right])$.

## 2. Main Result

Consider a binary code $C_{n}\subseteq {\left\{0,1\right\}}^{n}$ with fixed blocklength *n* and size *M* to be used over the memoryless BSC with crossover probability $0<p<\frac{1}{2}$. Denote the prior probability on $C_{n}$ by $P_{X^{n}}$ and hence $P_{X^{n}}\left(C_{n}\right)=1$. Without loss of generality, we assume that all codewords in $C_{n}$ occur with positive probability, i.e., $P_{X^{n}}\left(x^{n}\right)>0$ for all $x^{n}\in C_{n}$; hence, $C_{n}$ is the support of $P_{X^{n}}$.

It is known the minimal probability of decoding error is achieved by the MAP decoder, which upon the reception of the channel output $y^{n}\in {\left\{0,1\right\}}^{n}$ estimates the codeword $x^{n}\in C_{n}$ according to
(3)$$\begin{array}{c} e\left(y^{n}\right)=\arg\max_{u^{n}\in C_{n}}P_{X^{n}|Y^{n}}\left(u^{n}|y^{n}\right), \end{array}$$
where $P_{X^{n}|Y^{n}}$ is the posterior conditional distribution of $X^{n}$ given $Y^{n}$. We can see from (3) that if more than one $u^{n}\in C_{n}$ achieves the maximum value of $P_{X^{n}|Y^{n}}\left(u^{n}|y^{n}\right)$ for a given $y^{n}$, a decoder tie occurs, in which case the set of these $u^{n}$, denoted conveniently as $\left\{e\left(y^{n}\right)\right\}$, contains more than one element. As a result, an erroneous MAP decision is made if one of the two situations occurs: i) the transmitted codeword does not belong to $\left\{e\left(y^{n}\right)\right\}$; ii) the transmitted codeword belong to $\left\{e\left(y^{n}\right)\right\}$ and $|\{e\left(y^{n}\right)\}|>1$, but the tie breaker picks the wrong one from $\left\{e\left(y^{n}\right)\right\}$. By conveniently denoting
(4)$$C_{n}=\left\{c_{1},c_{2},\cdots ,c_{M}\right\},$$
the probability of the first situation acts as a lower bound $b_{n}$ for $a_{n}$ (i.e., $b_{n}\leq a_{n}$), where $b_{n}$ is given in (2) and can be written as
(5)$$\begin{array}{ccc} b_{n} & = & \sum_{i=1}^{M}P_{X^{n}}\left(c_{i}\right)\;P_{Y^{n}|X^{n}}\left(\left\{y^{n}\in {\left\{0,1\right\}}^{n}:P_{X^{n}|Y^{n}}\left(c_{i}|y^{n}\right)<\max_{r\in [M]\backslash \{i\}}P_{X^{n}|Y^{n}}\left(c_{r}|y^{n}\right)\right\}\right). \end{array}$$
It is shown in [22] that $b_{n}$ exactly equals the generalized Poor–Verdú (lower) bound [23,24] as its tilting parameter approaches infinity. The probability of the second situation is bounded above by the probability that the transmitted codeword belongs to $\left\{e\left(y^{n}\right)\right\}$ and $|\{e\left(y^{n}\right)\}|>1$, disregarding whether the tie breaker picks the wrong codeword or not, and this upper bound can be expressed as
(6)$$\begin{array}{ccc} \delta_{n} & ≜ & \sum_{i=1}^{M}P_{X^{n}}\left(c_{i}\right)\;P_{Y^{n}|X^{n}}\left(\left\{y^{n}\in {\left\{0,1\right\}}^{n}:P_{X^{n}|Y^{n}}\left(c_{i}|y^{n}\right)=\max_{r\in [M]\backslash \{i\}}P_{X^{n}|Y^{n}}\left(c_{r}|y^{n}\right)\right\}\right). \end{array}$$
We thus have
(7)$$b_{n}\leq a_{n}\leq b_{n}+\delta_{n}.$$
By proving the inequality
(8)$$\delta_{n}\leq 2qnb_{n},$$
where
(9)$$q≜\frac{1-p}{p}>1,$$
we have our main result as follows.

**Theorem** **2.**
*For any sequence of binary codes ${\left\{C_{n}\right\}}_{n\geq 1}$ and prior probabilities ${\left\{P_{X^{n}}\right\}}_{n\geq 1}$ used over the BSC, we have*

(10)
$$b_{n}\leq a_{n}\leq \left(1+2qn\right)b_{n}.$$



**Remark** **1.**
*Theorem 2 implies that the relative deviation of $a_{n}$ from $b_{n}$ is at most linear in the blocklength n and the impact of decoder ties in (6) to $a_{n}$ is only sub-exponential. Consequently, $a_{n}$ and $b_{n}$ must have the same error exponent. Note also that the upper bound in (10) differs from the result in Theorem 1 by an additional multiplicative factor of 2 in the qn term. As explained in the introduction section, this is a consequence of the fact that the probability of the group of channel output tuples that cause decoder ties but fail the flipping manipulation property is upper bounded by that of the remaining tie-inducing channel outputs. The full technical details are provided in Section 3.2. Finally, we emphasize that Theorem 2 holds for arbitrary binary codes, including “bad” codes for which high probability codewords have small Hamming distance between them. Hence, tightening the upper bound in (10) by restricting the analysis for “sufficiently good" codes, in the sense that their most likely codewords sit “sufficiently” far apart in ${\left\{0,1\right\}}^{n}$, is an interesting future direction.*


*List of Main Symbols*: Before providing an overview of the main steps of the proof of Theorem 2 (which is presented in full detail in the next section), we describe in Table 1 the main symbols used in the paper and indicate the equation where they are first introduced. We emphasize that sets $T_{j|i}$, $N_{j|i}$ and $S_{1,j}^{\left(m\right)}$ are defined *differently* from their counterparts in [1] that use the same notation.

We also visually illustrate in Figure 1 some of the main sets defined in Table 1 under the setting of Example 1, which is presented in Section 3 below for a non-uniformly distributed binary code with M=4 codewords and blocklength n=4 given by $C_{4}=\left\{c_{1},c_{2},c_{3},c_{4}\right\}=\left\{0000,0101,0110,0111\right\}$. More specifically, we only show the non-empty component subsets in $Y^{n}={\left\{0,1\right\}}^{4}$ corresponding to codewords $c_{1}$ and $c_{2}$; refer to Table A2 in Appendix A for a detailed listing of all component subsets in ${\left\{0,1\right\}}^{4}$ (including empty ones).

*Overview of the Proof*: Given that codeword $c_{i}$ is sent over the channel, i∈[M], let $T_{i}$ denote the set of output tuples $y^{n}$ that result in MAP decoding ties:(11)$$\begin{array}{ccc} T_{i} & ≜ & \left\{y^{n}\in {\left\{0,1\right\}}^{n}:P_{X^{n}|Y^{n}}\left(c_{i}|y^{n}\right)=\max_{r\in [M]\backslash \{i\}}P_{X^{n}|Y^{n}}\left(c_{r}|y^{n}\right)\right\} \end{array}$$(12)$$\begin{array}{ccc} \; & = & \left\{y^{n}\in {\left\{0,1\right\}}^{n}:P_{X^{n},Y^{n}}\left(c_{i},y^{n}\right)=\max_{r\in [M]\backslash \{i\}}P_{X^{n},Y^{n}}\left(c_{r},y^{n}\right)\right\}, \end{array}$$
where (12) holds because $P_{X^{n}|Y^{n}}\left(x^{n}|y^{n}\right)=\frac{P_{X^{n},Y^{n}}\left(x^{n},y^{n}\right)}{P_{Y^{n}}\left(y^{n}\right)}$. Then, $\delta_{n}$ in (6) can be rewritten as
(13)$$\delta_{n}=\sum_{i\in [M]}P_{X^{n}}\left(c_{i}\right)\;P_{Y^{n}|X^{n}}\left(T_{i}|c_{i}\right)=\sum_{i\in [M]}P_{X^{n},Y^{n}}\left(c_{i},T_{i}\right).$$
Similarly, let $N_{i}$ denote the set of output tuples $y^{n}$ which guarantee a tie-free MAP decoding error when $c_{i}$ is transmitted over the channel:(14)$$\begin{array}{ccc} N_{i} & ≜ & \left\{y^{n}\in {\left\{0,1\right\}}^{n}:P_{X^{n}|Y^{n}}\left(c_{i}|y^{n}\right)<\max_{r\in [M]\backslash \{i\}}P_{X^{n}|Y^{n}}\left(c_{r}|y^{n}\right)\right\} \end{array}$$(15)$$\begin{array}{ccc} \; & = & \left\{y^{n}\in {\left\{0,1\right\}}^{n}:P_{X^{n},Y^{n}}\left(c_{i},y^{n}\right)<\max_{r\in [M]\backslash \{i\}}P_{X^{n},Y^{n}}\left(c_{r},y^{n}\right)\right\}. \end{array}$$
Hence, $b_{n}$ in (5) can be rewritten as:(16)$$b_{n}=\sum_{i\in [M]}P_{X^{n}}\left(c_{i}\right)\;P_{Y^{n}|X^{n}}\left(N_{i}|c_{i}\right)=\sum_{i\in [M]}P_{X^{n},Y^{n}}\left(c_{i},N_{i}\right).$$
Note if $\delta_{n}=0$, then (7) is tight and (10) holds trivially; so, without loss of generality, we assume in the proof that $\delta_{n}>0$, which implies that there exists at least one non-empty $T_{i}$ for i∈[M]. Then, according to (13) and (16), we have that
(17)$$\begin{array}{ccc} \frac{\delta_{n}}{b_{n}} & = & \frac{\sum_{i\in [M]}P_{X^{n},Y^{n}}\left(c_{i},T_{i}\right)}{\sum_{i\in [M]}P_{X^{n},Y^{n}}\left(c_{i},N_{i}\right)}\leq \frac{\sum_{i\in \left[M\right]:T_{i}\neq \emptyset }P_{X^{n},Y^{n}}\left(c_{i},T_{i}\right)}{\sum_{i\in \left[M\right]:T_{i}\neq \emptyset }P_{X^{n},Y^{n}}\left(c_{i},N_{i}\right)}. \end{array}$$
We can upper-bound (17) by
(18)$$\frac{\sum_{i\in \left[M\right]:T_{i}\neq \emptyset }P_{X^{n},Y^{n}}\left(c_{i},T_{i}\right)}{\sum_{i\in \left[M\right]:T_{i}\neq \emptyset }P_{X^{n},Y^{n}}\left(c_{i},N_{i}\right)}\leq \max_{1\in \left[M\right]:T_{i}\neq \emptyset }\frac{P_{X^{n},Y^{n}}\left(c_{i},T_{i}\right)}{P_{X^{n},Y^{n}}\left(c_{i},N_{i}\right)},$$
where for convenience we will refer to an inequality of the form given in (18) as the *ratio-sum* inequality. As a result, Theorem 2 holds if we can substantiate that 2qn is an upper bound for (18). To this end, we will find a *proper partition* of $T_{i}$ and *an equal number* of disjoint subsets of $N_{i}$, of which the individual probabilities can be evaluated. For ease of notation, we denote the individual probabilities corresponding to the *K*-partition of $T_{i}$ and *K* disjoint subsets of $N_{i}$ by ${\left\{\alpha_{k}\right\}}_{k=1}^{K}$ and ${\left\{\beta_{k}\right\}}_{k=1}^{K}$, respectively. Then, we obtain that
(19)$$\begin{array}{c} \frac{P_{X^{n},Y^{n}}\left(c_{i},T_{i}\right)}{P_{X^{n},Y^{n}}\left(c_{i},N_{i}\right)}\leq \frac{\sum_{k=1}^{K}\alpha_{k}}{\sum_{k=1}^{K}\beta_{k}}. \end{array}$$
By showing that each individual ratio $\alpha_{k}/\beta_{k}$, k∈[K] is bounded above by 2qn, the ratio-sum inequality can again be applied to complete the proof.

## 3. Proof of Theorem 2

In [1], where a uniformly distributed prior probability $P_{X^{n}}$ over $C_{n}$ is assumed, one can flip a properly selected bit in the output $y^{n}\in T_{i}$ to convert it to a corresponding element in $N_{i}$. In light of this connection, one can evaluate the ratio $\frac{P_{Y^{n}|X^{n}}\left(T_{i}|c_{i}\right)}{P_{Y^{n}|X^{n}}\left(N_{i}|c_{i}\right)}$. This approach, however, no longer works when a non-uniformly distributed prior probability is considered. Therefore, we have to devise a more judicious approach to extend the result in [1] for a general prior probability.

### 3.1. A Partition of Non-Empty $T_{i}$ and Corresponding Disjoint Subsets of $N_{i}$

In this section, instead of finding a disjoint covering of the set of decoder ties $T_{i}$ as in [1], we establish a proper partition of $T_{i}$ from Definitions 1 and 2. This is one of the key differences from the techniques used in [1]. Example 1 is given after Proposition 1 to illustrate Definitions 1 and 2.

Given $y^{n}\in T_{i}$ defined in (12), there exists at least one m∈[M]\{i} such that
(20)$$\begin{array}{c} P_{X^{n},Y^{n}}\left(c_{i},y^{n}\right)=P_{X^{n},Y^{n}}\left(c_{m},y^{n}\right)=\max_{r\in [M]\backslash \{i\}}P_{X^{n},Y^{n}}\left(c_{r},y^{n}\right). \end{array}$$
We collect the indices *m* that satisfy (20) in $I_{i}\left(y^{n}\right)$ as follows:(21)$$I_{i}\left(y^{n}\right)≜\left\{h\in \left[M\right]\backslash \left\{i\right\}:P_{X^{n},Y^{n}}\left(c_{i},y^{n}\right)=P_{X^{n},Y^{n}}\left(c_{h},y^{n}\right)=\max_{r\in [M]\backslash \{i\}}P_{X^{n},Y^{n}}\left(c_{r},y^{n}\right)\right\}.$$

**Remark** **2.**
*First, we note that $I_{i}\left(y^{n}\right)$ is not empty as long as $y^{n}\in T_{i}$. Also, for any $y^{n}\in T_{i}$, we can infer from (21) that $h\in I_{i}\left(y^{n}\right)$ if and only if $y^{n}\in T_{h}$.*


In Definitions 1 and 2 that follow, we will assign each $y^{n}\in T_{i}$ to a subset indexed by $j\in I_{i}\left(y^{n}\right)$. These subsets will form a partition of $T_{i}$ as stated in Proposition 1.

**Definition** **1.**
*For j∈[M]\{i}, denoting by $S_{i,j}$ the set of indices where the bit components of $c_{i}$ and $c_{j}$ differ, we define*

{(22a)Tj|i≜yn∈Ti:j=minr∈Ii(yn):d(ci,yn|Si,r)<|Si,r|r;Nj|i≜{yn∈Ni:PXn,Yn(ci,yn)·q=PXn,Yn(cj,yn)·1q(22b)andPXn,Yn(cj,yn)≠PXn,Yn(cr,yn)forr∈[j−1]\{i}}.



Since there may exist $y^{n}\in T_{i}$ satisfying $d\left(c_{i},y^{n}|S_{i,r}\right)=\left|S_{i,r}\right|$ for all $r\in I_{i}\left(y^{n}\right)$, the collection of all elements in ${⋃}_{j\in [M]\backslash \{i\}}T_{j|i}$ may not exhaust the elements in $T_{i}$ (see Example 1). We thus go on to collect the remaining elements in $T_{i}\backslash {⋃}_{j\in [M]\backslash \{i\}}T_{j|i}$ as follows.

**Definition** **2.**
*Define for j∈[M]\{i},*

(23)
$$\begin{array}{ccc} T_{j|i} & ≜ & \left\{y^{n}\in T_{i}\;\backslash \;\left(\underset{h\in [M]\backslash \{i\}}{⋃}T_{h|i}\right):j=\min_{r\in I_{i}\left(y^{n}\right)}r\right\}. \end{array}$$



With the sets defined in Definitions 1 and 2, a partition of $T_{i}$ and disjoint subsets of $N_{i}$ are constructed as proven in the following proposition.

**Proposition** **1.**
*For non-empty $T_{i}$, the following two properties hold.*
*(i)* 
*The collection ${\left\{T_{j|i}⋃T_{j|i}\right\}}_{j\in [M]\backslash \{i\}}$ forms a (disjoint) partition of $T_{i}$.*
*(ii)* 
*${\left\{N_{j|i}\right\}}_{j\in [M]\backslash \{i\}}$ is a collection of disjoint subsets of $N_{i}$.*



Before proving Proposition 1, we provide the following example to illustrate the above sets.

**Example** **1.**
*This example illustrates the necessity of introducing $T_{j|i}$ as a companion to $T_{j|i}$. Suppose $C_{4}=\left\{c_{1},c_{2},c_{3},c_{4}\right\}=\left\{0000,0101,0110,0111\right\}$. Let $P_{X^{4}}\left(c_{1}\right)=\frac{q^{2}}{2+q^{2}+q^{-2}}$, $P_{X^{4}}\left(c_{2}\right)=P_{X^{4}}\left(c_{3}\right)=\frac{1}{2+q^{2}+q^{-2}}$ and $P_{X^{4}}\left(c_{4}\right)=\frac{q^{-2}}{2+q^{2}+q^{-2}}$. Then, $y^{4}=0111$ satisfies*

(24)
$$\underset{\frac{q^{2}}{\left(2+q^{2}+q^{-2}\right)}p^{4}q^{1}}{\underset{︸}{P_{X^{4},Y^{4}}\left(c_{1},y^{4}\right)}}=\max_{r\in [4]\backslash \{1\}}P_{X^{4},Y^{4}}\left(c_{r},y^{4}\right)=\underset{\frac{1}{\left(2+q^{2}+q^{-2}\right)}p^{4}q^{3}}{\underset{︸}{P_{X^{4},Y^{4}}\left(c_{2},y^{n}\right)}}=\underset{\frac{1}{\left(2+q^{2}+q^{-2}\right)}p^{4}q^{3}}{\underset{︸}{P_{X^{4},Y^{4}}\left(c_{3},y^{4}\right)}}>\underset{\frac{q^{-2}}{\left(2+q^{2}+q^{-2}\right)}p^{4}q^{4}}{\underset{︸}{P_{X^{4},Y^{4}}\left(c_{4},y^{4}\right)}},$$

*where the probabilities $P_{X^{n},Y^{n}}\left(x^{n},y^{n}\right)$ are written in the form*

(25)
$$P_{X^{n},Y^{n}}\left(x^{n},y^{n}\right)=P_{X^{n}}\left(x^{n}\right)P_{Y^{n}|X^{n}}\left(y^{n}|x^{n}\right)=P_{X^{n}}\left(x^{n}\right)p^{n}q^{n-d(x^{n},y^{n})}.$$


*Note that the first equality in (24) indicates $0111\in T_{1}$ and the last two equalities and the right-most inequality jointly imply $I_{1}\left(0111\right)=\left\{2,3\right\}$. In light of Proposition 1, this 0111 must lie in one and only one of ${\left\{T_{j|1}⋃T_{j|1}\right\}}_{j\in [4]\backslash \{1\}}$ as shown in Table A1 and Table A2 of Appendix A. Since there exist no integers h in $I_{1}\left(0111\right)$ fulfilling $d\left(c_{1},0111|S_{1,h}\right)<\left|S_{1,h}\right|$, this 0111 belongs to $T_{j|1}$ with $j=\min_{r\in I_{1}\left(0111\right)}=2$. Recall that in [1], an element in $N_{j|1}$ can be obtained if flipping a zero of $y^{n}\in T_{j|1}$ can make it further away from $c_{1}$ but closer to $c_{j}$. However, for $y^{4}=0111$ in this example if we flip the only zero to one, it gets further away from both $c_{1}$ and $c_{h}$ for any h=2,3,4. Therefore, the bit-flipping manipulation fails.*

*With $y^{4}=0111$, we also have*

(26)
$$\underset{\frac{1}{\left(2+q^{2}+q^{-2}\right)}p^{4}q^{3}}{\underset{︸}{P_{X^{4},Y^{4}}\left(c_{2},y^{4}\right)}}=\max_{r\in [4]\backslash \{2\}}P_{X^{4},Y^{4}}\left(c_{r},y^{4}\right)=\underset{\frac{q^{2}}{\left(2+q^{2}+q^{-2}\right)}p^{4}q}{\underset{︸}{P_{X^{4},Y^{4}}\left(c_{1},y^{4}\right)}}=\underset{\frac{1}{\left(2+q^{2}+q^{-2}\right)}p^{4}q^{3}}{\underset{︸}{P_{X^{4},Y^{4}}\left(c_{3},y^{n}\right)}}>\underset{\frac{1}{\left(2+q^{2}+q^{-2}\right)}p^{4}q^{4}}{\underset{︸}{P_{X^{4},Y^{4}}\left(c_{4},y^{4}\right)}},$$

*where the first equality indicates $0111\in T_{2}$ and the remaining parts in (26) jointly imply that $I_{2}\left(0111\right)=\left\{1,3\right\}$. Proposition 1 then states that this 0111 lies in one and only one of $\{T_{j|2}$$⋃T_{j|2}{\}}_{j\in [4]\backslash \{2\}}$. Since $1\in I_{2}\left(0111\right)=\left\{1,3\right\}$ and $d\left(c_{2},0111|S_{2,1}\right)=0<\left|S_{2,1}\right|=2$, we have $0111\in T_{1|2}$ according to (22a). Thus, we can flip a bit in 0111 to get further away from $c_{2}$ and closer to $c_{j}$ simultaneously. More specifically, the bit-flipping manipulation produces either 0110 or 0011, which lies in $N_{1|2}$ as $y^{4}=0111$ is in $T_{1|2}$. Therefore, we can associate the element in $T_{1|2}$ with an element in $N_{1|2}$ via a single flipping operation. For completeness, a full list of the sets $T_{i}$, $N_{i}$, $T_{j|i}$, $T_{j|i}$ and $N_{j|i}$ for i∈[4] and j∈[4]\{i}, is given in Appendix A.*


**Proof** **of Proposition 1.**First, we note that by the definitions in (22a) and (23), ${\left\{T_{j|i}\right\}}_{j\in [M]\backslash \{i\}}$ are disjoint and so is ${\left\{T_{j|i}\right\}}_{j\in [M]\backslash \{i\}}$. Additionally, (23) implies $T_{j|i}⋂T_{h|i}=\emptyset$ for arbitrary j,h∈[M]\{i}. Furthermore, according to Definitions 1 and 2, for any $y^{n}\in T_{i}$, we have either $y^{n}\in T_{h|i}$ or $y^{n}\in T_{h|i}$ for some h∈[M]\{i}. Consequently, ${\left\{T_{j|i}⋃T_{j|i}\right\}}_{j\in [M]\backslash \{i\}}$ forms a partition of $T_{i}$.On the other hand, the inequality in (22b) prevents multiple inclusions of an element from the previous collections. Therefore, ${\left\{N_{j|i}\right\}}_{j\in [M]\backslash \{i\}}$ are a collection of disjoint subsets of $N_{i}$. □

**Remark** **3.**
*When channel inputs are uniformly distributed as considered in [1], it follows that*

(27)
$$I_{i}\left(y^{n}\right)=\left\{h\in \left[M\right]\backslash \left\{i\right\}:d\left(c_{i},y^{n}\right)=d\left(c_{h},y^{n}\right)=\max_{r\in [M]\backslash \{i\}}d\left(c_{r},y^{n}\right)\right\},$$

*and $d\left(c_{i},y^{n}|S_{i,j}\right)=\frac{|S_{i,j}|}{2}<\left|S_{i,j}\right|$ for every $j\in I_{i}\left(y^{n}\right)$. Therefore, (22a) is reduced to*

(28)
$$\begin{array}{ccc} T_{j|i} & = & \left\{y^{n}\in T_{i}:j=\min_{r\in I_{i}\left(y^{n}\right)}r\right\}, \end{array}$$

*and*

(29)
$$\begin{array}{c} T_{j|i}=\emptyset . \end{array}$$


*We then have the following two remarks. First, we note that the $T_{j|i}$ newly defined via (22a) and reduced to (28) in the regime considered in [1] is more restrictive than the $T_{j|i}$ introduced in [1] (Equation (16a)). As a consequence, ${\left\{T_{j|i}\right\}}_{j\in [M]\backslash \{i\}}$ forms a partition of $T_{i}$ in this paper while those introduced in [1] (Equation (16a)) are a disjoint covering of $T_{i}$ under uniform channel inputs. Second, (29) shows that [1] does not need to consider a companion $T_{j|i}$ to $T_{j|i}$, but this paper does.*


Based on Proposition 1, we continue the derivation from (17) and obtain:(30)$$\begin{array}{ccc} \frac{\delta_{n}}{b_{n}} & \leq & \frac{\sum_{i\in [M]}P_{X^{n},Y^{n}}\left(c_{i},{⋃}_{j\in [M]\backslash \{i\}}\left(T_{j|i}⋃T_{j|i}\right)\right)}{\sum_{i\in [M]}P_{X^{n},Y^{n}}\left(c_{i},{⋃}_{j\in [M]\backslash \{i\}}N_{j|i}\right)} \end{array}$$(31)$$\begin{array}{ccc} & = & \frac{\sum_{i\in [M]}\sum_{j\in [M]\backslash \{i\}}P_{X^{n},Y^{n}}\left(c_{i},T_{j|i}\right)+\sum_{i\in [M]}\sum_{j\in [M]\backslash \{i\}}P_{X^{n},Y^{n}}\left(c_{i},T_{j|i}\right)}{\sum_{i\in [M]}\sum_{j\in [M]\backslash \{i\}}P_{X^{n},Y^{n}}\left(c_{i},N_{j|i}\right)}, \end{array}$$
where (31) holds because ${\left\{T_{j|i}\right\}}_{j\in [M]\backslash \{i\}}$ and ${\left\{T_{j|i}\right\}}_{j\in [M]\backslash \{i\}}$ are disjoint, and the same applies to $\{N_{j|i}$${\}}_{j\in [M]\backslash \{i\}}$. An additional upper bound for (31) requires the verification of the inequality:(32)$$\sum_{i\in [M]}\sum_{j\in [M]\backslash \{i\}}P_{X^{n},Y^{n}}\left(c_{i},T_{j|i}\right)\leq \sum_{j\in [M]}\sum_{i\in [M]\backslash \{j\}}P_{X^{n},Y^{n}}\left(c_{j},T_{i|j}\right),$$
which is an immediate consequence of the proposition to be proven in the next section (Proposition 2), stating that
(33)$$\begin{array}{ccc} y^{n} & \in & T_{j|i}\;\mathrm{and}\;h\in I_{i}\left(y^{n}\right)\Rightarrow y^{n}\in T_{\ell |h}\;\mathrm{for}\;\mathrm{some}\;\\ & & \;\;\;\;\;\;\;\;\;\;\;\;\;\;\;\;\;\ell \in I_{h}\left(y^{n}\right)\;\mathrm{and}\;P_{X^{n},Y^{n}}\left(c_{i},y^{n}\right)=P_{X^{n},Y^{n}}\left(c_{h},y^{n}\right). \end{array}$$

### 3.2. Verification of (32)

Recall that the main technique used in [1] is to associate every element in $T_{i}$ with a corresponding element in $N_{i}$ via the bit-flipping manipulation. By this bit-flipping association, the probability ratio of the elements and corresponding elements respectively in $T_{i}$ and $N_{i}$ can be evaluated. However, as Example 1 indicates, for an element in $T_{j|i}$, the bit-flipping association no longer works. This reveals the challenge of generalizing the results in [1] from uniform channel inputs to arbitrarily distributed channel inputs. A solution is to subdivide the elements in $T_{i}$ into two groups ${\left\{T_{j|i}\right\}}_{j\in [M]\backslash \{i\}}$ and ${\left\{T_{j|i}\right\}}_{j\in [M]\backslash \{i\}}$, where the bit-flipping association to ${\left\{N_{j|i}\right\}}_{j\in [M]\backslash \{i\}}$ works for the former group but not for the latter. The inequality in (32) can then be used to exclude the latter group with an upper bound:(34)$$\begin{array}{ccc} \frac{\delta_{n}}{b_{n}} & \leq & \frac{\sum_{i\in [M]}\sum_{j\in [M]\backslash \{i\}}P_{X^{n},Y^{n}}\left(c_{i},T_{j|i}\right)+\sum_{i\in [M]}\sum_{j\in [M]\backslash \{i\}}P_{X^{n},Y^{n}}\left(c_{i},T_{j|i}\right)}{\sum_{i\in [M]}\sum_{j\in [M]\backslash \{i\}}P_{X^{n},Y^{n}}\left(c_{i},N_{j|i}\right)} \end{array}$$(35)$$\begin{array}{ccc} & \;\leq & 2\frac{\sum_{i\in [M]}\sum_{j\in [M]\backslash \{i\}}P_{X^{n},Y^{n}}\left(c_{i},T_{j|i}\right)}{\sum_{i\in [M]}\sum_{j\in [M]\backslash \{i\}}P_{X^{n},Y^{n}}\left(c_{i},N_{j|i}\right)}. \end{array}$$

Since uniform channel inputs as considered in [1] guarantee (29), it can be seen from (35) that the multiplicative factor of 2 can be reduced to 1 as observed in Remark 1. For general arbitrary channel inputs, we have the factor of 2 since the set $T_{j|i}$ may not be empty. The validity of (32) can be confirmed by the next proposition.

**Proposition** **2.**
*Suppose $y^{n}\in T_{j|i}$. Then, for every $h\in I_{i}\left(y^{n}\right)$, we have*

(36)
$$\begin{array}{c} y^{n}\in T_{\ell |h}\;\mathrm{for}\;\mathrm{some}\;\ell \in I_{h}\left(y^{n}\right)\;\mathrm{and}\;P_{X^{n},Y^{n}}\left(c_{i},y^{n}\right)=P_{X^{n},Y^{n}}\left(c_{h},y^{n}\right). \end{array}$$



**Proof.** Suppose $y^{n}\in T_{j|i}$. Then, $d\left(c_{i},y^{n}|S_{i,h}\right)=\left|S_{i,h}\right|$ for every $h\in I_{i}\left(y^{n}\right)$. We therefore have:
(37)$$P_{X^{n},Y^{n}}\left(c_{i},y^{n}\right)=P_{X^{n},Y^{n}}\left(c_{h},y^{n}\right)=\max_{r\in [M]\backslash \{i\}}P_{X^{n},Y^{n}}\left(c_{r},y^{n}\right).$$We can rewrite (37) as
(38)$$P_{X^{n},Y^{n}}\left(c_{h},y^{n}\right)=P_{X^{n},Y^{n}}\left(c_{i},y^{n}\right)=\max_{r\in [M]\backslash \{h\}}P_{X^{n},Y^{n}}\left(c_{r},y^{n}\right),$$
implying $y^{n}\in T_{h}$ and $i\in I_{h}\left(y^{n}\right)$. Noting that $d\left(c_{h},y^{n}|S_{h,i}\right)=0<\left|S_{h,i}\right|$ because $d\left(c_{i},y^{n}|S_{i,h}\right)=\left|S_{i,h}\right|$ and $S_{h,i}=S_{i,h}$, we conclude that the smallest integer $\ell \in I_{h}\left(y^{n}\right)$ satisfying $d\left(c_{h},y^{n}|S_{h,\ell }\right)<\left|S_{h,\ell }\right|$ exists, and therefore $y^{n}\in T_{\ell |h}$. □

**Remark** **4.***Two observations can be made based on Proposition 2. First, Proposition 2 indicates that every $y^{n}\in T_{j|i}$ must appear at least once in the sum $\sum_{h\in [M]}$$\sum_{\ell \in [M]\backslash \{h\}}$
$P_{X^{n},Y^{n}}\left(c_{h},T_{\ell |h}\right)$, contributing the same probability mass $P_{X^{n},Y^{n}}\left(c_{h},y^{n}\right)$ as $P_{X^{n},Y^{n}}\left(c_{i},y^{n}\right)$. Second, Proposition 2 also implies that every $y^{n}\in T_{j|i}$ cannot be contained in $\left({⋃}_{h\in [M]}{⋃}_{r\in [M]\backslash \{h\}}T_{r|h}\right)\backslash T_{j|i}$. This observation can be substantiated as follows. For every $h\in I_{i}\left(y^{n}\right)$, Proposition 2 implies $y^{n}\in T_{\ell |h}$ for some $\ell \in I_{h}\left(y^{n}\right)$ and hence Definition 2 immediately gives $y^{n}\notin T_{r|h}$ for all r∈[M]\{h}. For $h\notin I_{i}\left(y^{n}\right)$, we have $y^{n}\notin T_{h}$ and therefore $y^{n}\notin T_{r|h}\subseteq T_{h}$ for all r∈[M]\{h} as pointed out in Remark 2. As a result, every $y^{n}\in T_{j|i}$ appears *exactly* once in the sum $\sum_{h\in [M]}$$\sum_{r\in [M]\backslash \{h\}}P_{X^{n},Y^{n}}\left(c_{h},T_{r|h}\right)$. Combining the two observations leads to:*(39)$$\begin{array}{ccc} \sum_{i\in [M]}\sum_{j\in [M]\backslash \{i\}}\sum_{y^{n}\in T_{j|i}}P_{X^{n},Y^{n}}\left(c_{i},y^{n}\right) & \leq & \sum_{j\in [M]}\sum_{\ell \in [M]\backslash \{j\}}\sum_{y^{n}\in T_{\ell |j}}P_{X^{n},Y^{n}}\left(c_{j},y^{n}\right). \end{array}$$

To flesh out the above inequality, we give the next example.

**Example** **2.**
*Proceeding from Example 1, we observe from Table A1 and Table A2 in Appendix A that 0111 is contained in $T_{2|1}$, $T_{1|2}$ and $T_{1|3}$. Hence, it appears once in the sum $\sum_{j\in [4]}\sum_{i\in [4]\backslash \{j\}}$
$P_{X^{n},Y^{n}}\left(c_{j},T_{i|j}\right)$ while it contributes twice in the sum $\sum_{j\in [4]}\sum_{i\in [4]\backslash \{j\}}P_{X^{n},Y^{n}}\left(c_{j},T_{i|j}\right)$. We then confirm from (A35) that:*

(40)
$$\begin{array}{ccc} \sum_{i\in [4]}\sum_{j\in [4]\backslash \{i\}}P_{X^{4},Y^{4}}\left(c_{i},T_{j|i}\right) & \geq & \sum_{i\in [4]}\sum_{j\in [4]\backslash \{i\}}P_{X^{4},Y^{4}}\left(c_{i},T_{j|i}\right). \end{array}$$



We continue the derivation from (35) and obtain
(41)$$\begin{array}{ccc} \frac{\delta_{n}}{b_{n}} & \leq & 2\frac{\sum_{i\in [M]}\sum_{j\in \left[M\right]\backslash \left\{i\right\}:T_{j|i}\neq \emptyset }P_{X^{n},Y^{n}}\left(c_{i},T_{j|i}\right)}{\sum_{i\in [M]}\sum_{j\in \left[M\right]\backslash \left\{i\right\}:T_{j|i}\neq \emptyset }P_{X^{n},Y^{n}}\left(c_{i},N_{j|i}\right)} \end{array}$$
(42)$$\begin{array}{ccc} & \leq & 2\max_{i\in \left[M\right]\;\mathrm{and}\;j\in \left[M\right]\backslash \left\{i\right\}:T_{j|i}\neq \emptyset }\frac{P_{X^{n},Y^{n}}\left(c_{i},T_{j|i}\right)}{P_{X^{n},Y^{n}}\left(c_{i},N_{j|i}\right)}, \end{array}$$
where we add the restriction $T_{j|i}\neq \emptyset$ in (41) to exclude the cases of zero dividing by zero in (42), and (42) follows the ratio-sum inequality in (18).

In the next section, we introduce a number of delicate decompositions of non-empty $T_{j|i}$ and an equal number of disjoint subsets of $N_{j|i}$ to facilitate the bit-flipping association of the pairs.

### 3.3. Atomic Decomposition of
Non-Empty $T_{j|i}$ and the Corresponding Disjoint Subsets of $N_{j|i}$

To simplify the exposition, we assume without loss of generality that $c_{1}$ is the all-zero codeword (It is known that we can simultaneously flip the same position of all codewords to yield a new code of equal performance over the BSC. Thus, via a number of flipping manipulations, we can transform any code to a code of equal performance with the first codeword being all-zero.) Below we present the proof for i=1. The proof for general i>1 follows annalagously.

Since $c_{1}$ is the all-zero codeword, $S_{1,j}$ is the set containing the indices of the non-zero components of $c_{j}$. To facilitate the investigation of the structure of $c_{j}$ relative to the remaining codewords ${\left\{c_{r}\right\}}_{r\in [M]\backslash \{1,j\}}$, we first partition $S_{1,j}$ into $2^{M-2}$ subsets according to whether each index in $S_{1,j}$ is in $S_{1,2}$, *…*, $S_{1,j-1}$, $S_{1,j+1}$, *…*, $S_{1,M}$ or not, as follows:(43)$$S_{1,j}^{\left(m\right)}≜\left(\overset{j-1}{\underset{r=2}{⋂}}S_{r;\lambda_{r}}\right)⋂\left(\overset{M}{\underset{r=j+1}{⋂}}S_{r;\lambda_{r}}\right)⋂S_{1,j}\;\mathrm{for}\;m≜1+\sum_{r=2}^{j-1}\lambda_{r}\cdot 2^{r-2}+\sum_{r=j+1}^{M}\lambda_{r}\cdot 2^{r-3},$$
where $S_{r;1}≜S_{1,r}$ and $S_{r;0}≜\left[n\right]\backslash S_{1,r}=S_{1,r}^{c}$, and each $\lambda_{r}\in \left\{0,1\right\}$. An example of the partition is given below.

**Example** **3.**
*Suppose $C_{4}=\left\{00000,11001,01111,01101\right\}$. For j=3 and $S_{1,j}=\left\{2,3,4,5\right\}$, we obtain $2^{4-2}=4$ subsets as*

(44)
$$\begin{array}{c} S_{1,3}^{\left(m\right)}=\left\{\begin{array}{cc} S_{1,3}^{\left(1\right)}=S_{1,2}^{c}⋂S_{1,4}^{c}⋂S_{1,3}=\left\{4\right\}, & \mathrm{if}\;\left(\lambda_{4},\lambda_{2}\right)=\left(0,0\right);\\ S_{1,3}^{\left(2\right)}=S_{1,2}⋂S_{1,4}^{c}⋂S_{1,3}=\emptyset , & \mathrm{if}\;\left(\lambda_{4},\lambda_{2}\right)=\left(0,1\right);\\ S_{1,3}^{\left(3\right)}=S_{1,2}^{c}⋂S_{1,4}⋂S_{1,3}=\left\{3\right\}, & \mathrm{if}\;\left(\lambda_{4},\lambda_{2}\right)=\left(1,0\right);\\ S_{1,3}^{\left(4\right)}=S_{1,2}⋂S_{1,4}⋂S_{1,3}=\left\{2,5\right\}, & \mathrm{if}\;\left(\lambda_{4},\lambda_{2}\right)=\left(1,1\right). \end{array}\right. \end{array}$$



As $c_{1}$ is the all-zero codeword, the components of $c_{r}$ with indices in $S_{1,j}^{\left(m\right)}$ can now be unambiguously identified and must all equal $\lambda_{r}$. As a result,
(45)$$d\left(c_{1},c_{r}|S_{1,j}^{\left(m\right)}\right)=\left\{\begin{array}{cc} |S_{1,j}^{\left(m\right)}|, & \lambda_{r}=1;\\ 0, & \lambda_{r}=0. \end{array}\right.$$

**Example** **4.**
*Proceeding from Example 3, we have*

(46)
$$\begin{array}{c} \left\{\begin{array}{cc} d\left(c_{1},c_{2}|S_{1,3}^{\left(1\right)}\right)=0 & \mathrm{because}\;\lambda_{2}=0;\\ d\left(c_{1},c_{2}|S_{1,3}^{\left(2\right)}\right)=\left|S_{1,3}^{\left(2\right)}\right|=0 & \mathrm{because}\;\lambda_{2}=1;\\ d\left(c_{1},c_{2}|S_{1,3}^{\left(3\right)}\right)=0 & \mathrm{because}\;\lambda_{2}=0;\\ d\left(c_{1},c_{2}|S_{1,3}^{\left(4\right)}\right)=\left|S_{1,3}^{\left(4\right)}\right|=2 & \mathrm{because}\;\lambda_{2}=1, \end{array}\right. \end{array}$$

*and*

(47)
$$\begin{array}{c} \left\{\begin{array}{cc} d\left(c_{1},c_{4}|S_{1,3}^{\left(1\right)}\right)=0 & \mathrm{because}\;\lambda_{4}=0;\\ d\left(c_{1},c_{4}|S_{1,3}^{\left(2\right)}\right)=0 & \mathrm{because}\;\lambda_{4}=0;\\ d\left(c_{1},c_{4}|S_{1,3}^{\left(3\right)}\right)=\left|S_{1,3}^{\left(3\right)}\right|=1 & \mathrm{because}\;\lambda_{4}=1;\\ d\left(c_{1},c_{4}|S_{1,3}^{\left(4\right)}\right)=\left|S_{1,3}^{\left(4\right)}\right|=2 & \mathrm{because}\;\lambda_{4}=1. \end{array}\right. \end{array}$$



It should be emphasized that $S_{1,j}^{\left(m\right)}$ in this paper is defined differently from that in [1]. While the one defined in [1] partitions $S_{1,j}$ only according to codewords with indices less than *j*, the one defined in this paper considers all other M−2 codewords in the partition manipulation, and hence the order of codewords becomes irrelevant.

Next, to decompose $T_{j|1}$, we further define a sequence of incremental sets:(48)$$S_{1,j}^{\left(m\right)}≜\overset{m}{\underset{h=1}{⋃}}S_{1,j}^{\left(h\right)},\;m\in \left[2^{M-2}\right],$$
and set $S_{1,j}^{\left(0\right)}≜\emptyset$. Let $\ell_{1,j}≜\left|S_{1,j}\right|$ and $\ell_{1,j}^{\left(m\right)}≜\left|S_{1,j}^{\left(m\right)}\right|$ respectively denote the sizes of $S_{1,j}$ and $S_{1,j}^{\left(m\right)}$ and note that $0=\ell_{1,j}^{\left(0\right)}\leq \ell_{1,j}^{\left(1\right)}\leq \ell_{1,j}^{\left(2\right)}\leq \cdots \leq \ell_{1,j}^{\left(2^{M-2}\right)}=\ell_{1,j}$.

The idea behind the partition of $T_{j|1}$ into $\ell_{1,j}$ subsets, indexed by $k\in \left[\ell_{1,j}-1\right]⋃\left\{0\right\}$, is as follows. Pick one $y^{n}\in T_{j|1}$. We start by examining whether $d(c_{1},y^{n}|S_{1,j}^{\left(1\right)})$ is strickly less than $\ell_{1,j}^{\left(1\right)}-1$. If the answer is negative, we continue examining whether $d(c_{1},y^{n}|S_{1,j}^{\left(2\right)})$ is strictly less than $\ell_{1,j}^{\left(2\right)}-1$. Proceed until we reach the smallest *m* such that $d\left(c_{1},y^{n}|S_{1,j}^{\left(m\right)}\right)<\ell_{1,j}^{\left(m\right)}-1$ holds. Setting *k* to be equal to $k=d(c_{1},y^{n}|S_{1,j}^{\left(m\right)})$, we assign this $y^{n}$ to the subset $T_{j|1}\left(k\right)$. Notably, there exists no such number $m\in [2^{M-2}]$ that satisfies $d\left(c_{1},y^{n}|S_{1,j}^{\left(m\right)}\right)<\ell_{1,j}^{\left(m\right)}-1$ if and only if $d\left(c_{1},y^{n}|S_{1,j}\right)=\ell_{1,j}-1$; in this case, we find the smallest *m* satisfying $S_{1,j}^{\left(m\right)}=S_{1,j}$ and assign this element to $T_{j|1}\left(\ell_{1,j}-1\right)$ as $d\left(c_{1},y^{n}|S_{1,j}^{\left(m\right)}\right)=\ell_{1,j}-1$. For ease of describing the above algorithmic partition process, we introduce a mapping from $k\in \left[\ell_{1,j}-1\right]⋃\left\{0\right\}$ to $m\in [2^{M-2}]$ as follows:(49)$$\eta_{k}≜\left\{\begin{array}{cc} \min\left\{m\in \left[2^{M-2}\right]:k<\ell_{1,j}^{\left(m\right)}-1\right\}, & 0\leq k<\ell_{1,j}-1;\\ \min\left\{m\in \left[2^{M-2}\right]:k=\ell_{1,j}^{\left(m\right)}-1\right\}, & k=\ell_{1,j}-1. \end{array}\right.$$
We can see that for $0\leq k<\ell_{1,j}-1$, we have $\ell_{1,j}^{\left(\eta_{k}-1\right)}-1\leq k<\ell_{1,j}^{\left(\eta_{k}\right)}-1$. Therefore, if $y^{n}\in T_{j|1}$ is assigned to $T_{j|1}\left(k\right)$ for some $k<\ell_{1,j}-1$, we must have
(50)$$\ell_{1,j}^{\left(\eta_{k}-1\right)}-1\leq d\left(c_{1},y^{n}|S_{1,j}^{\left(\eta_{k}-1\right)}\right)\leq d\left(c_{1},y^{n}|S_{1,j}^{\left(\eta_{k}\right)}\right)=k<\ell_{1,j}^{\left(\eta_{k}\right)}-1.$$
On the other hand, if $y^{n}\in T_{j|1}$ is collected in $T_{j|1}\left(\ell_{1,j}-1\right)$, then $S_{1,j}^{\left(\eta_{k}\right)}=S_{1,j}$ and
(51)$$\ell_{1,j}^{\left(\eta_{k}-1\right)}-1\leq d\left(c_{1},y^{n}|S_{1,j}^{\left(\eta_{k}-1\right)}\right)\leq d\left(c_{1},y^{n}|S_{1,j}^{\left(\eta_{k}\right)}\right)=\ell_{1,j}-1.$$
A formal definition of $T_{j|1}\left(k\right)$ is given next, where the corresponding subsets $N_{j|1}\left(k\right)$ of $N_{j|1}$ are also introduced.

**Definition** **3.**
*Define for k=0, *1*, …, $\ell_{1,j}-1$,*

{(52a)Tj|1(k)≜yn∈Tj|1:ℓ1,j(ηk−1)−1≤dc1,yn|S1,j(ηk−1)anddc1,yn|S1,j(ηk)=k;(52b)Nj|1(k)≜yn∈Nj|1:ℓ1,j(ηk−1)=dc1,yn|S1,j(ηk−1)anddc1,yn|S1,j(ηk)=k+1,

*where $\eta_{k}$ is defined in (49).*


With Definition 3, we have the following proposition.

**Proposition** **3.**
*For non-empty $T_{j|1}$, the following two properties hold.*
*(i)* 
*${\left\{T_{j|1}\left(k\right)\right\}}_{k\in \left[\ell_{1,j}-1\right]⋃\left\{0\right\}}$ forms a partition of $T_{j|1}$;*
*(ii)* 
*${\left\{N_{j|1}\left(k\right)\right\}}_{k\in \left[\ell_{1,j}-1\right]⋃\left\{0\right\}}$ is a collection of disjoint subsets of $N_{j|1}$.*



**Proof.** It can be seen from the definitions of ${\left\{T_{j|1}\left(k\right)\right\}}_{k\in \left[\ell_{1,j}-1\right]\;⋃\left\{0\right\}}$ and ${\left\{N_{j|1}\left(k\right)\right\}}_{k\in \left[\ell_{1,j}-1\right]\;⋃\left\{0\right\}}$ that they are collections of mutually disjoint subsets of $T_{j|1}$ and $N_{j|1}$, respectively. It remains to argue that every element in $T_{j|1}$ belongs to $T_{j|1}\left(k\right)$ for some $k\in \left[\ell_{1,j}-1\right]⋃\left\{0\right\}$. Noting that the element $y^{n}$ in $T_{j|1}$ satisfies $d\left(c_{1},y^{n}|S_{1,j}\right)\leq \ell_{1,j}-1$, we differentiate two cases: $d\left(c_{1},y^{n}|S_{1,j}\right)\leq \ell_{1,j}-2$ and $d\left(c_{1},y^{n}|S_{1,j}\right)=\ell_{1,j}-1$. For the former case, $d\left(c_{1},y^{n}|S_{1,j}^{\left(m\right)}\right)<\ell_{1,j}^{\left(m\right)}-1$ must hold for $m=\eta_{k}$; hence, this $y^{n}$ will be contained in $T_{j|1}\left(k\right)$. For the latter case, $y^{n}$ will be included in $T_{j|1}\left(\ell_{1,j}-1\right)$. The proof is thus completed. □

In light of Proposition 3, we can apply the ratio-sum inequality to obtain
(53)$$\begin{array}{ccc} \frac{P_{X^{n},Y^{n}}\left(c_{1},T_{j|1}\right)}{P_{X^{n},Y^{n}}\left(c_{1},N_{j|1}\right)} & \leq & \frac{\sum_{k=0:T_{j|1}\left(k\right)\neq \emptyset }^{\ell_{1,j-1}}P_{X^{n},Y^{n}}\left(c_{1},T_{j|1}\left(k\right)\right)}{\sum_{k=0:T_{j|1}\left(k\right)\neq \emptyset }^{\ell_{1,j-1}}P_{X^{n},Y^{n}}\left(c_{1},N_{j|1}\left(k\right)\right)} \end{array}$$
(54)$$\begin{array}{ccc} & \leq & \max_{k\in \left[\ell_{1,j}-1\right]⋃\left\{0\right\}:T_{j|1}\left(k\right)\neq \emptyset }\frac{P_{X^{n},Y^{n}}\left(c_{1},T_{j|1}\left(k\right)\right)}{P_{X^{n},Y^{n}}\left(c_{1},N_{j|1}\left(k\right)\right)}. \end{array}$$

We continue to construct a fine partition of $T_{j|1}\left(k\right)$ and the corresponding disjoint subsets of $N_{j|1}\left(k\right)$ in Proposition 4 after giving the next definition.

**Definition** **4.**
*Define for $u^{n}\in T_{j|1}\left(k\right)$,*

{(55a)Tj|1(un;k)≜yn∈Tj|1(k):dun,yn|S1,j(ηk)c=0;(55b)Nj|1(un;k)≜yn∈Nj|1(k):dun,yn|(S1,j(ηk))c=0,

*where $\eta_{k}$ is given in (49).*


Note from Definition 4 that for one element $u^{n}$ in non-empty $T_{j|1}\left(k\right)$, we can find a group of elements that have identical bit components to $u^{n}$ with indices in ${\left(S_{1,j}^{\left(\eta_{k}\right)}\right)}^{c}$. We denote this group as $T_{j|1}\left(u^{n};k\right)$. We continue this grouping manipulation until all elements in $T_{j|1}\left(k\right)$ are exhausted as summarized below.

**Proposition** **4.**
*For non-empty $T_{j|1}\left(k\right)$, there exists a representative subset $U_{j|1}\left(k\right)\subseteq T_{j|1}\left(k\right)$ such that the following two properties hold.*
*(i)* 
*${\left\{T_{j|1}\left(u^{n};k\right)\right\}}_{u^{n}\in U_{j|1}\left(k\right)}$ forms a (non-empty) partition of $T_{j|1}\left(k\right)$;*
*(ii)* 
*${\left\{N_{j|1}\left(u^{n};k\right)\right\}}_{u^{n}\in U_{j|1}\left(k\right)}$ is a collection of (non-empty) disjoint subsets of $N_{j|1}\left(k\right)$.*



Since the above proposition can be self-validated via its sequential selection manipulation of each $u^{n}$ from $T_{j|1}\left(k\right)$, we omit the proof. Interested readers can find the details in [1] (Section III-C).

From Proposition 4, using again the ratio-sum inequality, we obtain that for non-empty $T_{j|1}\left(k\right)$,
(56)$$\begin{array}{ccc} \frac{P_{X^{n},Y^{n}}\left(c_{1},T_{j|1}\left(k\right)\right)}{P_{X^{n},Y^{n}}\left(c_{1},N_{j|1}\left(k\right)\right)} & \leq & \frac{\sum_{u^{n}\in U_{j|1}\left(k\right)}P_{X^{n},Y^{n}}\left(c_{1},T_{j|1}\left(u^{n};k\right)\right)}{\sum_{u^{n}\in U_{j|1}\left(k\right)}P_{X^{n},Y^{n}}\left(c_{1},N_{j|1}\left(u^{n};k\right)\right)} \end{array}$$
(57)$$\begin{array}{ccc} & \leq & \max_{u^{n}\in U_{j|1}\left(k\right)}\frac{P_{X^{n},Y^{n}}\left(c_{1},T_{j|1}\left(u^{n};k\right)\right)}{P_{X^{n},Y^{n}}\left(c_{1},N_{j|1}\left(u^{n};k\right)\right)}. \end{array}$$

Noting that the above result can be similarly conducted for general i>1, we combine (42), (54) and (57) to conclude that
(58)$$\begin{array}{c} \frac{\delta_{n}}{b_{n}}\leq 2\max_{i\in \left[M\right]\;\mathrm{and}\;j\in \left[M\right]\backslash \left\{i\right\}:T_{j|i}\neq \emptyset }\max_{k\in \left[\ell_{i,j}-1\right]⋃\left\{0\right\}:T_{j|i}\left(k\right)\neq \emptyset }\max_{u^{n}\in U_{j|i}\left(k\right)}\frac{P_{X^{n},Y^{n}}\left(c_{i},T_{j|i}\left(u^{n};k\right)\right)}{P_{X^{n},Y^{n}}\left(c_{i},N_{j|i}\left(u^{n};k\right)\right)}. \end{array}$$
The final task is to evaluate $P_{X^{n},Y^{n}}\left(c_{i},T_{j|i}\left(u^{n};k\right)\right)/P_{X^{n},Y^{n}}\left(c_{i},N_{j|i}\left(u^{n};k\right)\right)$ in order to characterize a linear upper bound for $\delta_{n}/b_{n}$.

### 3.4. Characterization of a Linear Upper Bound for $\delta_{n}/b_{n}$

We again focus on i=1 with $c_{1}$ being the all-zero codeword for simplicity. The definitions of $T_{j|1}\left(u^{n};k\right)$ in (55a) and $N_{j|1}\left(u^{n};k\right)$ in (55b) indicate that when dealing with the ratio $P_{X^{n},Y^{n}}\left(c_{1},T_{j|1}\left(u^{n};k\right)\right)/P_{X^{n},Y^{n}}\left(c_{1},N_{j|1}\left(u^{n};k\right)\right)$, we only need to consider those bits with indices in $S_{1,j}^{\left(\eta_{k}\right)}$ because the remaining bits of all tuples in $T_{j|1}\left(u^{n};k\right)$ and $N_{j|1}\left(u^{n};k\right)$ have identical values as $u^{n}$. Note that all $|T_{j|1}\left(u^{n};k\right)|$ elements in $T_{j|1}\left(u^{n};k\right)$ have exactly *k* ones with indices in $S_{1,j}^{\left(\eta_{k}\right)}$, and all $|N_{j|1}\left(u^{n};k\right)|$ elements in $N_{j|1}\left(u^{n};k\right)$ have exactly k+1 ones with indices in $S_{1,j}^{\left(\eta_{k}\right)}$, we can immediately infer that
(59)$$\frac{P_{X^{n},Y^{n}}\left(c_{1},T_{j|1}\left(u^{n};k\right)\right)}{P_{X^{n},Y^{n}}\left(c_{1},N_{j|1}\left(u^{n};k\right)\right)}=\frac{P_{X^{n}}\left(c_{1}\right)\cdot P_{Y^{n}|c_{1}}\left(T_{j|1}\left(u^{n};k\right)|c_{1}\right)}{P_{X^{n}}\left(c_{1}\right)\cdot P_{Y^{n}|c_{1}}\left(N_{j|1}\left(u^{n};k\right)|c_{1}\right)}=\frac{\left(1-p\right)}{p}\cdot \frac{|T_{j|1}\left(u^{n};k\right)|}{|N_{j|1}\left(u^{n};k\right)|}.$$
The cardinalities of $T_{j|1}\left(u^{n};k\right)$ and $N_{j|1}\left(u^{n};k\right)$ then decide the ratio in (59) as verified in the next proposition, based on which the proof of Theorem 2 can be completed from (58).

**Proposition** **5.**
*For $u^{n}\in T_{j|1}\left(k\right)$, we have*

(60)
$$\frac{P_{X^{n},Y^{n}}\left(c_{1},T_{j|1}\left(u^{n};k\right)\right)}{P_{X^{n},Y^{n}}\left(c_{1},N_{j|1}\left(u^{n};k\right)\right)}\leq \frac{\left(1-p\right)}{p}n.$$



**Proof.** Recall from (22a), (52a) and (55a) that $y^{n}\in T_{j|1}\left(u^{n};k\right)\subseteq T_{j|1}\left(k\right)\subseteq T_{j|1}$ if and only if
{(61a)PXn,Yn(c1,yn)=PXn,Yn(cj,yn)=maxh∈[M]\{1}PXn,Yn(x(h)n,yn)andd(c1,yn|S1,j)<|Si,j|;(61b)ℓ1,j(ηk−1)−1≤dc1,yn|S1,j(ηk−1)anddc1,yn|S1,j(ηk)=k;(61c)dun,yn|(S1,j(ηk))c=0.
Thus, the number of elements in $T_{j|1}\left(u^{n};k\right)$ is exactly the number of channel outputs $y^{n}$ fulfilling the above three conditions. We then examine the number of $y^{n}$ satisfying (61b) and (61c). Noting that these $y^{n}$ have either $\ell_{1,j}^{\left(\eta_{k}-1\right)}-1$ ones or $\ell_{1,j}^{\left(\eta_{k}-1\right)}$ ones with indices in $S_{1,j}^{\left(\eta_{k}-1\right)}$, we know that there are at most
(62)ℓ1,j(ηk−1)ℓ1,j(ηk−1)−1ℓj(ηk)−ℓ1,j(ηk−1)k−(ℓ1,j(ηk−1)−1)+ℓ1,j(ηk−1)ℓ1,j(ηk−1)ℓj(ηk)−ℓ1,j(ηk−1)k−ℓ1,j(ηk−1)
of $y^{n}$ tuples satisfying (61b) and (61c). Disregarding (61a), we get that the number of elements in $T_{j|1}\left(u^{n};k\right)$ is upper-bounded by (62).On the other hand, from (22b), (52b) and (55b), we obtain that $w^{n}\in N_{j|1}\left(u^{n};k\right)\subseteq N_{j|1}\left(k\right)\subseteq N_{j|1}$ if and only if
{(63a)PXn,Yn(c1,wn)·q2=PXn,Yn(cj,wn);(63b)PXn,Yn(c1,wn)·q2≠PXn,Yn(cr,wn)forr∈[j−1]\{1};(63c)ℓ1,j(ηk−1)=dc1,wn|S1,j(ηk−1)anddc1,wn|S1,j(ηk)=k+1;(63d)dun,wn|(S1,j(ηk))c=0.
We then claim that any $w^{n}$ satisfying (63c) and (63d) directly validate (63a) and (63b). Note that the validity of the claim, which we prove in Appendix B, immediately implies that the number of elements in $N_{j|1}\left(u^{n};k\right)$ can be determined by (63c) and (63d), and hence
(64)|Nj|1(un;k)|=ℓj(ηk)−ℓ1,j(ηk−1)k+1−ℓ1,j(ηk−1).Under this claim, (62) and (64) result in
(65)|Tj|1(un;k)||Nj|1(un;k)|≤ℓ1,j(ηk−1)ℓ1,j(ηk−1)−1ℓj(ηk)−ℓ1,j(ηk−1)k−(ℓ1,j(ηk−1)−1)+ℓ1,j(ηk−1)ℓ1,j(ηk−1)ℓj(ηk)−ℓ1,j(ηk−1)k−ℓ1,j(ηk−1)ℓj(ηk)−ℓ1,j(ηk−1)k+1−ℓ1,j(ηk−1)
(66)$$\begin{array}{ccc} & = & \ell_{1,j}^{\left(\eta_{k}-1\right)}+\frac{k+1-\ell_{1,j}^{\left(\eta_{k}-1\right)}}{\ell_{j}^{\left(\eta_{k}\right)}-k} \end{array}$$
(67)$$\begin{array}{ccc} & \leq & \ell_{1,j}^{\left(\eta_{k}-1\right)}+\frac{\ell_{j}^{\left(\eta_{k}\right)}-\ell_{1,j}^{\left(\eta_{k}-1\right)}}{1} \end{array}$$
(68)$$\begin{array}{ccc} & \leq & \;n, \end{array}$$
where (67) holds because $k\leq \ell_{j}^{\left(\eta_{k}\right)}-1$ by (49), and (68) follows from $\ell_{1,j}^{\left(\eta_{k}\right)}\leq \ell_{1,j}\leq n$. The proof of the proposition is thus completed by (59) and (68). □

## 4. Conclusions

In this paper, we analyzed the error probability of block codes sent over the memoryless BSC under an arbitrary (not necessarily uniform) input distribution and used in conjunction with (optimal) MAP decoding. We showed that decoder ties do not affect the error exponent of the probability of error, thus extending a similar result recently established in [1] for uniformly distributed channel inputs. This result was obtained by proving that the relative deviation of the error probability from the probability of error grows no more than linearly in blocklength when no MAP decoding ties occur, directly implying that decoder ties have only a sub-exponential effect on the error probability as blocklength grows without bound. Future work includes further extending this result for more general channels used under arbitrary input statistics, such as non-binary symmetric channels (Note that the result of Theorem 1 can be extended for non-binary (*q*-ary, q>2) codes sent over *q*-ary symmetric memoryless channels under a uniform input distribution; see [25] (Theorem 2).) and binary non-symmetric channels. Studying how to sharpen the upper bound derived in (10) for “sufficiently good” codes as highlighted in Remark 1 and for codes with small blocklengths are other worthwhile future directions.

## Figures and Tables

**Figure 1 entropy-25-00668-f001:**
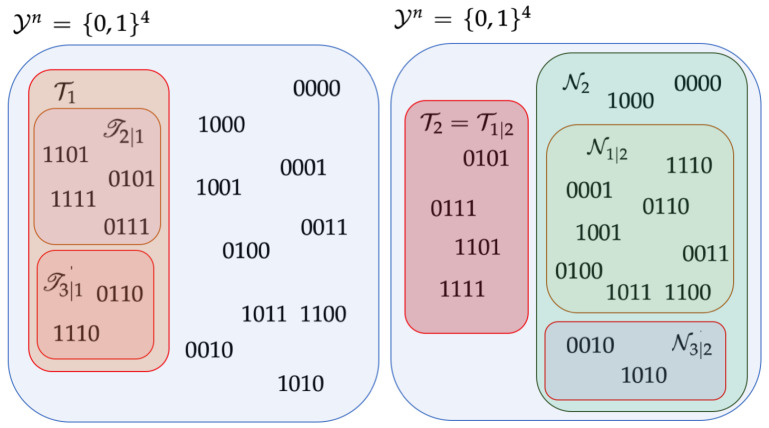
An illustration, based on the setting in Example 1 for a non-uniformly distributed binary code (with M=n=4) given by $C_{4}=\left\{c_{1},c_{2},c_{3},c_{4}\right\}=\left\{0000,0101,0110,0111\right\}$ of the non-empty component subsets of $Y^{n}$ defined in Table 1 and corresponding to codewords $c_{1}=0000$ (**left** figure) and $c_{2}=0101$ (**right** figure).

**Table 1 entropy-25-00668-t001:** Summary of the main symbols used in this paper.

Symbol	Description	Defined in
[M]	A shorthand for {1,2,…,M}	
$C_{n}$	The code $\left\{c_{1},c_{2},\ldots ,c_{n}\right\}$ with $c_{1}$ being the all-zero codeword	
$d(u^{n},v^{n}|S)$	The Hamming distance between the portions of $u^{n}$ and $v^{n}$ with indices in S	
*All terms below are functions of $C_{n}$ (this dependence is not explicitly shown to simplify notation)*		
$T_{i}$	The set of channel outputs $y^{n}$ inducing a decoder tie when $c_{i}$ is sent	(12)
$N_{i}$	The set of channel outputs $y^{n}$ leading to a tie-free decoder decision error when $c_{i}$ is sent	(15)
$I_{i}\left(y^{n}\right)$	The set $\left\{m\in \left[M\right]\backslash \left\{i\right\}:y^{n}\in T_{m}\right\}$ for $y^{n}\in T_{i}$	(21)
$S_{i,j}$	The set of indices for which the components of $c_{i}$ and $c_{j}$ differ	
$\ell_{i,j}$	The size of $S_{i,j}$, i.e., $|S_{i,j}|$	
$T_{j|i}$	The subset of $T_{i}$ consisting of channel outputs $y^{n}$ such that *j* is the minimal	(22a)
	number *r* in $I_{i}\left(y^{n}\right)$ satisfying $d\left(c_{i},y^{n}|S_{i,r}\right)<\ell_{i,r}$	
$N_{j|i}$	The subset of $N_{i}$ consisting of channel outputs $y^{n}$ that satisfy $P_{X^{n},Y^{n}}\left(c_{i},y^{n}\right)\cdot q=$	(22b)
	$P_{X^{n},Y^{n}}\left(c_{j},y^{n}\right)\cdot \frac{1}{q}$ and that are not included in $N_{r|i}$ for r∈[j−1]⊂{i}	
$T_{j|i}$	The subset of $T_{i}\backslash \left({⋃}_{h\in [M]\backslash \{i\}}T_{h|i}\right)$ consisting of channel outputs $y^{n}$	(23)
	such that *j* is the minimal number in $I_{i}\left(y^{n}\right)$	
$S_{1,j}^{\left(m\right)}$	The subset of $S_{1,j}$ defined according to whether each index in $S_{1,j}$ is in each	(43)
	of $S_{1,2}$, *…*, $S_{1,j-1}$, $S_{1,j+1}$, *…*, $S_{1,M}$	
$S_{1,j}^{\left(m\right)}$	The union of $S_{1,j}^{\left(1\right)}$, $S_{1,j}^{\left(2\right)}$, *…*, $S_{1,j}^{\left(m\right)}$	(48)
$\ell_{1,j}^{\left(m\right)}$	The size of $S_{1,j}^{\left(m\right)}$, i.e., $|S_{1,j}^{\left(m\right)}|$	
$\eta_{k}$	The mapping from $k\in \{0,1,\ldots ,\ell_{j}-1\}$ to $\left[2^{M-2}\right]$ used for partitioning $T_{j|1}$ into $\ell_{1,j}$	(49)
	subsets ${\left\{T_{j|1}\left(k\right)\right\}}_{0\leq k<\ell_{1,j}}$	
$T_{j|1}\left(k\right)$	The *k*th partition of $T_{j|1}$ for k=0, 1, *…*, $\ell_{1,j}-1$	(52a)
$N_{j|1}\left(k\right)$	The *k*th subset of $N_{j|1}$ for k=0, 1, *…*, $\ell_{1,j}-1$	(52b)
$U_{j|1}\left(k\right)$	The set of representative elements in $T_{j|1}\left(k\right)$ for partitioning $T_{j|1}\left(k\right)$	
$T_{j|1}\left(u^{n};k\right)$	The subset of $T_{j|1}\left(k\right)$ associated with $u^{n}\in U_{j|1}\left(k\right)$	(55a)
$N_{j|1}\left(u^{n};k\right)$	The subset of $N_{j|1}\left(k\right)$ associated with $u^{n}\in U_{j|1}\left(k\right)$	(55b)

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
