# Peer review of "On Decoder Ties for the Binary Symmetric Channel with Arbitrarily Distributed Input"

_entropy, 2023, doi:10.3390/e25040668_

Round 1

Reviewer 1 Report

See attached pdf.

Reviewer 2 Report

The topic of this paper is interesting and the math formulations seem rigorous but the reviewer has some concerns,

·         The reviewer suggest to add more details regarding the SoA. As is it too summarized and almost half of the introduction is dedicated to [1].

·         What is the motivation to have “Overview of the Proof” in section 2 and then an entire section 3 for the proof?

·         The paper is a little bit hard to follow since there are a lot of math formulation and proofs See the possibility to move some parts to the Annex.

·         Is it possible to add 1 or 2 figures with the results to make the paper more appealing?

Round 2

Reviewer 1 Report

Authors had addressed my (otherwise minor) comments.

Reviewer 2 Report

The authors have addressed the main reviewer concerns.